# Silencing of STE20-type kinase STK25 in human aortic endothelial and smooth muscle cells is atheroprotective

Emmelie Cansby [1], Sima Kumari[1], Mara Caputo[1], Ying Xia[1], Rando Porosk[2], Jonathan Robinson[3], Hao Wang [3], Britt-Marie Olsson[4], Josefine Vallin[1], Julie Grantham[1], Ursel Soomets [2], L. Thomas Svensson[3], Carina Sihlbom[4], Hanns-Ulrich Marschall [5], Andreas Edsfeldt[6,7,8], Isabel Goncalves[6,7] & Margit Mahlapuu [1]✉

Recent studies highlight the importance of lipotoxic damage in aortic cells as the major pathogenetic contributor to atherosclerotic disease. Since the STE20-type kinase STK25 has been shown to exacerbate ectopic lipid storage and associated cell injury in several metabolic organs, we here investigate its role in the main cell types of vasculature. We depleted STK25 by small interfering RNA in human aortic endothelial and smooth muscle cells exposed to oleic acid and oxidized LDL. In both cell types, the silencing of STK25 reduces lipid accumulation and suppresses activation of inflammatory and fibrotic pathways as well as lowering oxidative and endoplasmic reticulum stress. Notably, in smooth muscle cells, STK25 inactivation hinders the shift from a contractile to a synthetic phenotype. Together, we provide several lines of evidence that antagonizing STK25 signaling in human aortic endo-thelial and smooth muscle cells is atheroprotective, highlighting this kinase as a new potential therapeutic target for atherosclerotic disease.

[1] Department of Chemistry and Molecular Biology, University of Gothenburg and Sahlgrenska University Hospital, Gothenburg, Sweden. [2] Department of Biochemistry, Institute of Biomedicine and Translational Medicine, University of Tartu, Tartu, Estonia. [3] Department of Biology and Biological Engineering, National Bioinformatics Infrastructure Sweden, Science for Life Laboratory, Chalmers University of Technology, Gothenburg, Sweden. [4] Proteomics Core Facility, University of Gothenburg, Gothenburg, Sweden. [5] Department of Molecular and Clinical Medicine/Wallenberg Laboratory, Institute of Medicine, University of Gothenburg and Sahlgrenska University Hospital, Gothenburg, Sweden. [6] Department of Clinical Sciences Malmö, Clinical Research Center, Lund University, Malmö, Sweden. [7] Department of Cardiology, Skåne University Hospital, Lund/Malmö, Sweden. [8] Wallenberg Center for Molecular Medicine, Lund University, Malmö, Sweden. ✉email: Margit.Mahlapuu@gu.se

Cardiovascular diseases (CVDs) collectively comprise the number one cause of death globally[1,2]. The main pathological process of most CVDs is atherosclerosis[3], which results from the accumulation of lipids in the arterial intima, triggering the infiltration of inflammatory cells and, consequently, fueling oxidative and endoplasmic reticulum (ER) stress[4]. Many expected that the remarkable success of statins would halt the epidemic of atherosclerotic CVDs[5]. Yet, by 2030, about 24 million people are predicted to die from CVDs annually[6]. This huge and still growing burden of CVDs on individuals, families, and health-care systems indicates an urgent need for additional research efforts directed on understanding the molecular pathogenesis of atherosclerotic diseases and developing improved measures for their prevention as well as treatment.

Our translational studies in well-characterized patient cohorts, cultured human cells, and genetically modified mouse models, have identified serine/threonine protein kinase (STK)25, a member of the sterile 20 (STE20) kinase superfamily[7], as a critical regulator of lipid accumulation, inflammatory infiltration, as well as oxidative and ER stress in key metabolic tissues including liver, kidneys, and adipose depots[8–19]. Importantly, our recent research also reveals that STK25 controls atherosclerosis susceptibility in a mouse model of hypercholesterolemia[20]. In this study, we induced atherosclerosis in *Stk25* knockout and transgenic mice, and their wild-type littermates, by gene transfer of gain-of-function mutant of proprotein convertase subtilisin/kexin type 9 (PCSK9), which induces the downregulation of hepatic low-density lipoprotein receptor (LDLR), combined with a western-type diet. We found that the genetic ablation of STK25 effectively attenuates the formation of atherosclerosis lesions in this model independently from alterations in circulating lipid levels[20]. Reciprocally, *Stk25* transgenic mice present aggravated lipid accumulation in the aortic sinus and increased plaque maturation compared with wild-type littermates despite similar levels of fasting plasma cholesterol[20].

Although our studies using global depletion and overexpression of STK25 in knockout and transgenic mice, respectively, suggest a key role of STK25 in atherosclerotic plaque progression[20], it remains unknown whether the impact of this protein on atherosclerosis susceptibility is due to a direct effect of STK25 in aortic cells or a secondary action in other tissues. To decipher the cell-autonomous role of STK25 in the control of atherosclerosis, we here characterized the effects of STK25 inactivation in human primary aortic endothelial and smooth muscle cells. We found that STK25 cell-autonomously regulates ectopic lipid accumulation, activation of inflammatory and fibrotic pathways, and stimulation of oxidative and ER stress in both vascular cell types. Furthermore, STK25 was shown to control calcium deposition and the shift from a contractile to a synthetic (proliferative) phenotype in aortic smooth muscle cells. Together, STK25 emerges as a critical node governing the differences between human atheroprotective *vs.* atheroprone endothelial and smooth muscle cells.

## Results

**Silencing of STK25 suppresses lipid accumulation in human aortic endothelial and smooth muscle cells.** Lipid accumulation in aortic cells is known to trigger the initiation and progression of atherogenesis[21,22]. To explore the impact of STK25 knockdown on intracellular lipid storage in vascular cells, we transfected human aortic smooth muscle cells (hASMCs) and human aortic endothelial cells (hAECs) with *STK25*-specific small interfering (si) RNA or with a non-targeting control (NTC) siRNA (Supplementary Fig. 1). In all experiments described below, cells were studied under basal culture conditions as well as after treatment with oleic acid only or with both oleic acid and oxidized low-density lipoprotein (oxLDL). Cells transfected with *STK25* siRNA displayed substantially lower target mRNA and protein expression assessed by qRT-PCR and Western blot, respectively (Fig. 1a, b, h, i). Consistently, we detected no immunostaining using STK25 antibodies in cells transfected with *STK25* siRNA (Fig. 1c, j).

First, we stained hASMCs and hAECs with Bodipy 493/503, which detects neutral lipids within lipid droplets. We found that the Bodipy-positive area was about 2-fold lower in hAECs transfected with *STK25* siRNA compared with NTC siRNA both under basal conditions and after exposure to oleic acid alone or in combination with oxLDL (Fig. 1k). Similar difference was seen in STK25-deficient hASMCs after metabolic challenge; however, no significant reduction in Bodipy staining was observed in these cells under basal conditions (Fig. 1d). Next, we examined the mechanisms underlying the reduced lipid storage observed in hASMCs and hAECs by STK25 silencing. We found that the depletion of STK25 resulted in an increase in β-oxidation assessed by quantification of [$^3$H]-labeled water as the product of [9,10-$^3$H(N)]palmitic acid oxidation, both with and without metabolic challenge (Fig. 1e, l). We also revealed that fatty acid influx and triacylglycerol (TAG) synthesis were significantly lower in hASMCs and hAECs transfected with *STK25* siRNA compared with NTC siRNA, when exposed to oleic acid in combination with oxLDL but not under basal conditions (Fig. 1f, g, m, n). Additionally, we found that TAG hydrolase activity measured using [$^3$H]triolein as the substrate was higher in both cell types transfected with *STK25* siRNA *vs.* NTC siRNA after metabolic challenge (Supplementary Fig. 2).

**Knockdown of STK25 suppresses inflammation and fibrosis in human aortic endothelial and smooth muscle cells.** We found that in parallel to suppressed lipid deposition in STK25-deficient hASMCs and hAECs, the nuclear abundance of NFκB was less evident in cells transfected with *STK25* siRNA *vs.* NTC siRNA when challenged with oleic acid and oxLDL, indicating a lower degree of activation of inflammatory pathways (Fig. 2a, b, g, h). Additionally, the concentration of MCP-1 and IL-1β in the culture medium was reduced in STK25-deficient hASMCs and hAECs (Fig. 2c, i). The levels of IL-8 were significantly decreased by the depletion of STK25 only in hASMCs, although there was a similar tendency in hAECs, and no difference in TNFα was detected (Fig. 2c, i). MCP-1, IL-1β, and IL-8 are known to play an important role in the pathogenesis of atherosclerosis by stimulating the attraction and adhesion of monocytes[23–25]. Consistently, we found that knockdown of STK25 attenuated THP-1 monocyte adhesion in both hASMCs and hAECs when assessed after metabolic challenge (Fig. 2d, j).

To examine the effect of STK25 on aortic fibrosis, hASMCs and hAECs were incubated with TGF-β1 for 24 hours. Knockdown of STK25 had a significant impact on the TGF-β1-induced expression of several key markers of fibrosis in hAECs, both with and without metabolic challenge. The effect of STK25 depletion was less pronounced in hASMC, where only the *ELN* expression was found to be lower (Fig. 2e, k). Furthermore, after exposure to oleic acid and oxLDL, the production of soluble collagen was significantly reduced in both cell types transfected with *STK25* siRNA (Fig. 2f, l).

**STK25 controls aortic oxidative and ER stress in vitro and in vivo.** Excessive lipid storage in aortic cells is known to cause oxidative and ER stress, aggravating atherosclerosis progression[4,26]. To this end, we found that lower intracellular lipid deposition in STK25-deficient hASMCs and hAECs was accompanied by reduced oxidative stress as evidenced by markedly decreased levels

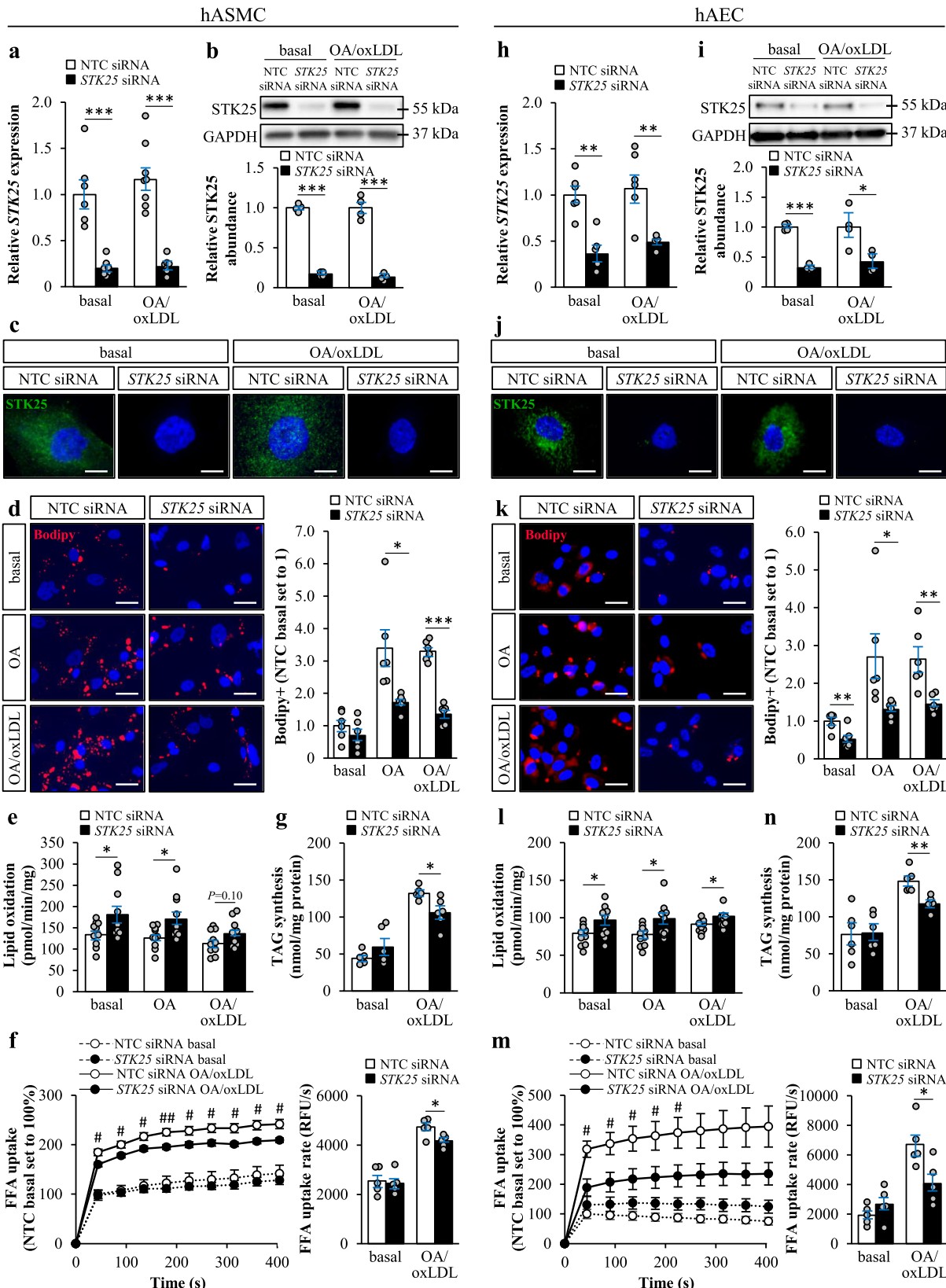

of superoxide radicals $(O_2^{\bullet-})$ quantified by dihydroethidium (DHE) staining, abrogated oxidative DNA damage detected by immunostaining for 8-oxoguanine (8-oxoG), and protection against oxidized phospholipids measured by immunostaining for E06, when assessed after exposure to oleic acid and oxLDL (Fig. 3a–c). In line with immunostainings, we found that hASMCs and hAECs

transfected with *STK25* siRNA displayed significantly lower reactive oxygen species (ROS) content when quantified using an ELISA kit, both with and without metabolic challenge (Fig. 3d, f). Oxidative stress is known to contribute to aortic cell dysfunction primarily through rapid inactivation of the key vasoprotective factor nitric oxide (NO)[4]. Here, lower ROS levels were accompanied by higher

**Fig. 1 Silencing of STK25 decreases lipid accumulation in human aortic endothelial and smooth muscle cells.** hASMCs and hAECs were transfected with *STK25* siRNA or NTC siRNA. **a, b, h, i** STK25 mRNA (**a, h**) and protein (**b, i**) abundance. Protein levels were analyzed by densitometry; representative Western blots are shown with glyceraldehyde-3-phosphate dehydrogenase (GAPDH) used as a loading control. **c, j** Representative immunofluorescence images of cells stained with anti-STK25 antibody (green); nuclei stained with DAPI (blue). **d, k** Representative images of cells stained with Bodipy (red); nuclei stained with DAPI (blue). Quantification of the staining. **e, l** Oxidation of radiolabeled palmitate. **f, m** Fatty acid uptake. **g, n** TAG synthesis from [14C]-labeled oleic acid. In (**c, j d, k**), the scale bars represent 10 and 25 μm, respectively. Data are mean ± SEM from 4-12 wells per group. FFA, free fatty acid; OA, oleic acid. *$P < 0.05$, **$P < 0.01$, ***$P < 0.001$ for *STK25* siRNA *vs.* NTC siRNA; #$P < 0.05$, ##$P < 0.01$ for *STK25* siRNA *vs.* NTC siRNA after exposure to oleic acid and oxLDL.

cellular NO content in hASMCs; however, no alterations were detected in hAECs (Fig. 3e, g).

In parallel with alterations in oxidative stress markers, we detected reduced immunostaining for KDEL (a signal motif for ER retrieval) and CHOP (a marker for ER stress-induced cell death) in STK25-depleted hASMCs and hAECs compared with cells transfected with NTC siRNA, after metabolic challenge (Fig. 4a, b). Additionally, gene expression of key markers of ER stress was significantly lower in both cell types after knockdown of STK25 (Fig. 4c).

We also quantified the levels of oxidative and ER stress markers in cross sections of the aortic sinus from *Stk25* knockout and transgenic mice, and their wild-type littermates, where atherosclerosis was induced by gene transfer of gain-of-function mutant of PCSK9 (hPCSK9[D374Y]) combined with an atherogenic western-type diet[20]. Consistent with our in vitro observations in hASMCs and hAECs, we found about 3-fold decrease in the abundance of 8-oxoG and E06 positive cells in the plaques from *Stk25*[−/−] *vs.* wild-type mice (Fig. 5a, b). Reciprocally, hPCSK9[D374Y]-transduced *Stk25* transgenic mice showed increased 8-oxoG and E06 positive area (Fig. 5a, b). Furthermore, aortic root sections from *Stk25*[−/−] mice showed reduced levels of ER stress indicators KDEL and CHOP compared with their wild-type littermates, and the opposite was observed in *Stk25* transgenic mice (Fig. 5c, d). These data complement and expand on our previously reported observations that *Stk25* knockout and transgenic mice display significantly smaller or larger atherosclerotic lesion area (Oil Red O-positive), respectively, compared with wild-type controls[20]. Notably, we also detected suppressed lipid storage as well as lower levels of oxidative and ER stress markers in primary mouse aortic smooth muscle cells (mASMCs) and mouse aortic endothelial cells (mAECs) transfected with *Stk25* siRNA *vs.* NTC siRNA after metabolic challenge, providing further support for a cell-autonomous mechanism underlying the atheroprotective effect observed in *Stk25*[−/−] mice (Supplementary Fig. 3a-d).

**Silencing of STK25 decreases phenotypic switching in human aortic smooth muscle cells.** During atherosclerosis progression, smooth muscle cells are known to shift from a contractile to a synthetic phenotype, which results in a gradual shifting from a differentiated to a dedifferentiated state, facilitating proliferation and migration[27,28]. To evaluate a possible effect of STK25 on proliferation, we compared the Ki67 positive area, as well as bromodeoxyuridine (BrdU) incorporation during DNA synthesis, in hASMCs and hAECs transfected with *STK25* siRNA *vs.* NTC siRNA. In hASMCs, knockdown of STK25 resulted in both lower levels of Ki67 and reduced DNA synthesis, demonstrating a decreased proliferation rate (Fig. 6a, b). Interestingly, no difference in Ki67 positive area or BrdU incorporation was detected in hAECs (Supplementary Fig. 4a, b). Next, we measured migration of hASMCs 6, 12, and 24 hours post-seeding on transwell inserts. We found that the number of migrated cells was diminished by knockdown of STK25 at all time points (Fig. 6c). Further, we detected markedly higher protein levels of several contractile

markers characterizing differentiated smooth muscle cells in STK25-deficient hASMCs [a several-fold increase in SMMHC (MYH11), SM22α, and calponin, while no change was observed in myocardin or LMOD1; Supplementary Fig. 5].

Importantly, synthetic smooth muscle cells are known to secrete extracellular vesicles that are critical for vascular calcification[29,30]. Calcification is an unfavorable event in the natural history of atherosclerosis, where initial calcium deposition in response to pro-inflammatory stimuli results in the formation of spotty or granular calcification, which in turn fuels further inflammation[31]. Here, we found a marked decrease in calcification in STK25-depleted hASMCs, both with and without metabolic challenge (Fig. 6d, Supplementary Fig. 6). We also detected lower mRNA expression of the calcification markers osteopontin (*SPP1*) and *RUNX2* in STK25-deficient hASMCs (Fig. 6e), which was accompanied by a tendency for reduced protein abundance (Fig. 6f). Notably, we did not detect any differences in calcium deposition in hAECs (Supplementary Fig. 4c).

Interestingly we found that the gene expression of several markers of endothelial cell dysfunction was altered in STK25-deficient hAECs. The mRNA abundance of KLF4 (Kruppel-like factor 4), one of the most important markers of an atheroprotective endothelium[32], was increased in hAECs transfected with *STK25* siRNA *vs.* NTC siRNA (Supplementary Fig. 7). Reciprocally, the transcript level of the proatherogenic cell surface receptors VCAM-1 and ICAM-1 were decreased in STK25-depleted hAECs (Supplementary Fig. 7).

To identify metabolic pathways affected by STK25, we performed metabolite profiling on hASMCs and hAECs after challenge with oleic acid and oxLDL. In hASMCs, the depletion of STK25 resulted in significantly higher levels of putrescine, which was not observed in STK25-deficient hAECs (Supplementary Fig. 8). Of note, it has recently been demonstrated that in macrophages putrescine augments the clearance, or efferocytosis, of apoptotic cells, ultimately contributing to the resolution of atherosclerosis[33]. The capacity of engulfment of apoptotic cells has also been reported in vascular smooth muscle cells[34]; however, it is not known whether putrescine mediates efferocytosis in this cell type. Consistent for both hASMCs and hAECs, the ratio of spermidine to putrescine was reduced about 2-fold in cells transfected with *STK25* siRNA *vs.* NTC siRNA, indicating lower activity of spermidine synthase (Supplementary Fig. 8). Moreover, in hASMCs, the levels and ratios of several other polyamines were also different upon STK25-depletion (Supplementary Fig. 8). Changes in the polyamine pathway have been previously linked to CVDs; however, it is still unclear which alterations associate with atheroprotective *vs.* dysfunctional phenotypes[35,36].

**Global phosphoproteomics analysis of STK25-deficient human aortic smooth muscle cells.** To decipher the molecular mechanisms by which STK25 hinders the transition of aortic smooth muscle cells to an atheroprone state, we performed global quantitative phosphoproteomic analysis by a liquid chromatography (LC)-

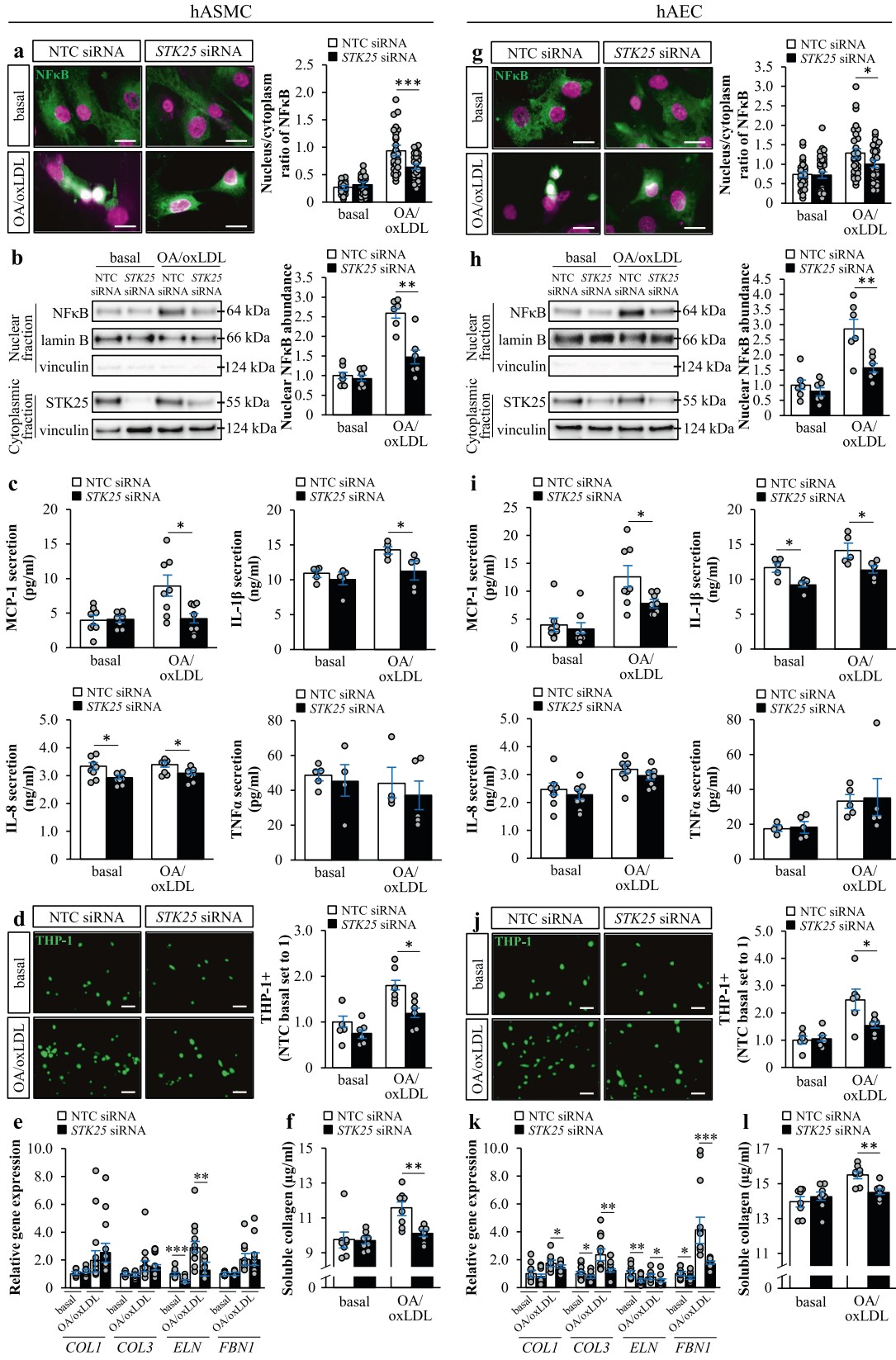

mass spectrometry (MS) technique on hASMCs transfected with *STK25* siRNA or NTC siRNA after metabolic challenge (Fig. 7a). We observed that the abundance of 121 peptides representing 30 unique proteins was differentially regulated by a factor of 1.4-fold or more, and with a *P* value below 0.05, in cells transfected with *STK25* siRNA *vs.* NTC siRNA when treated with oleic acid and oxLDL

(Fig. 7b, c, Supplementary Table 2). The analysis of phosphopeptide levels, after correction to their corresponding total protein levels, identified 37 peptides with significantly different phosphorylation status in STK25-depleted hASMCs. Out of these, the abundance of 21 phosphopeptides was reduced, representing possible target sites for the kinase activity of STK25 (Fig. 7d, e, Supplementary Table 3).

**Fig. 2 Knockdown of STK25 suppresses inflammation and fibrosis in human aortic endothelial and smooth muscle cells.** hASMCs and hAECs were transfected with *STK25* siRNA or NTC siRNA. **a**, **g** Representative immunofluorescence images of cells stained with anti-NFκB antibody (green); nuclei stained with DAPI (violet); colocalization shown in white. Quantification of the ratio of nuclear to cytoplasmic staining. **b**, **h** Nuclear extracts were analyzed by Western blot using antibody specific for NFκB. Protein levels were analyzed by densitometry; representative Western blots are shown with lamin B (nuclear marker) and vinculin (cytoplasmic marker) used as controls. Cytoplasmic fraction was analyzed by Western blot using antibody specific for STK25 to demonstrate the transfection efficacy; representative Western blots are shown with vinculin used as a loading control. **c**, **i** Concentration of secreted cytokines and chemokines. **d**, **j** Representative images of calcein acetoxymethyl-stained THP-1 monocytes (green). Quantification of the staining. **e**, **k** Relative gene expression of selected fibrosis markers 24 h after TGF-β1 stimulation. **f**, **l** Concentration of collagens in media 24 h after TGF-β1 stimulation. In (**a**, **g**) and (**d**, **j**), the scale bars represent 10 and 50 μm, respectively. Data in (**a**, **g**) are mean ± SEM from 30 cells per group; data in (**b**–**f**, **h**–**l**) are mean ± SEM from 4-12 wells per group. OA, oleic acid. *P < 0.05, **P < 0.01, ***P < 0.001 for *STK25* siRNA *vs.* NTC siRNA.

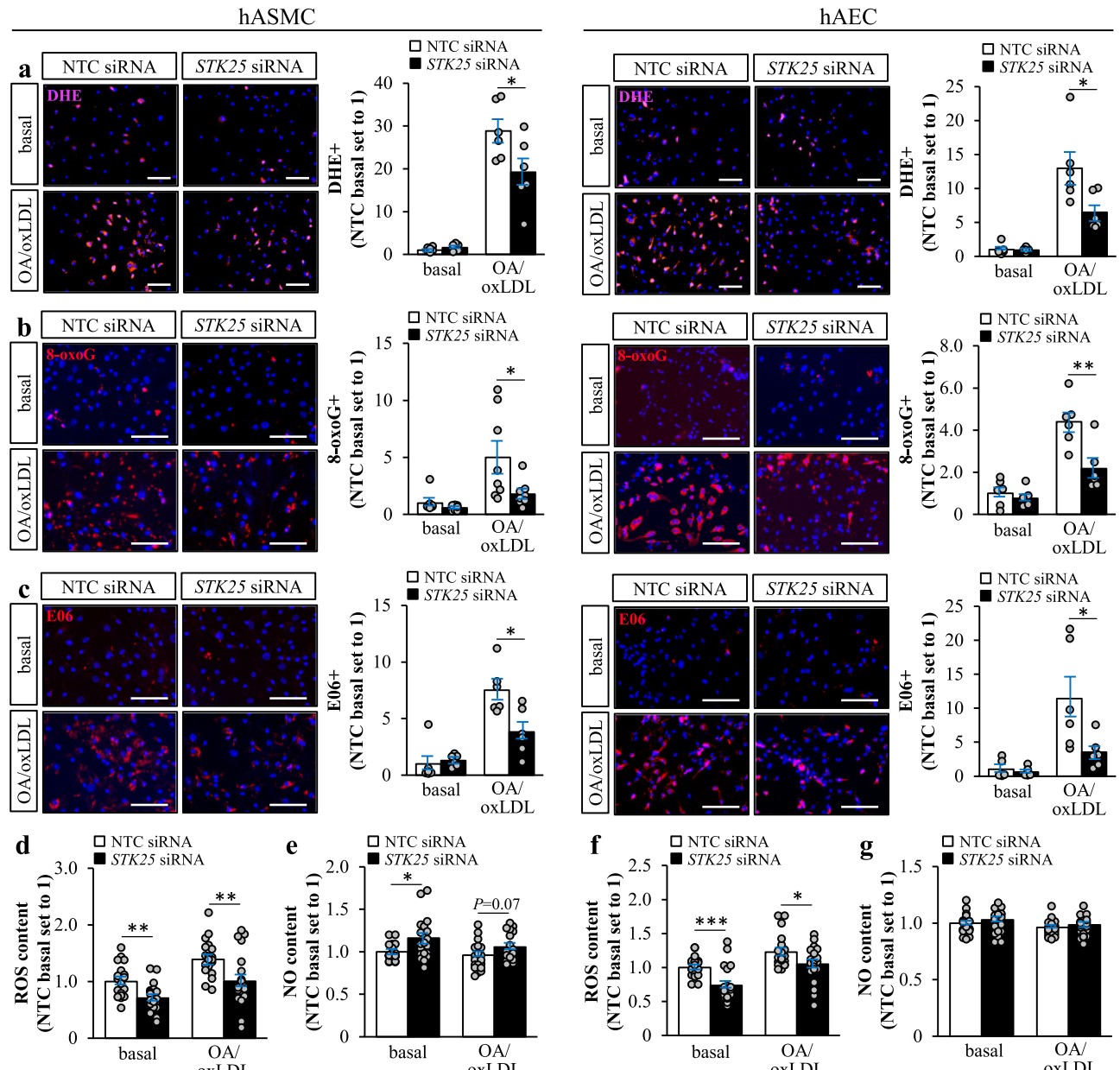

**Fig. 3 STK25 regulates oxidative stress in human aortic endothelial and smooth muscle cells.** hASMCs and hAECs were transfected with *STK25* siRNA or NTC siRNA. **a**–**c** Representative images of cells stained with DHE (pink) or processed for immunofluorescence with anti-8-oxoG (red) or anti-E06 (red) antibodies; nuclei stained with DAPI (blue). Quantification of the staining. **d**, **f** Cellular ROS content. **e**, **g** Cellular NO content. In (**a**–**c**), the scale bars represent 100 μm. Data in (**a**–**c**) are mean ± SEM from 6-8 wells per group; data in (**d**–**g**) are mean ± SEM from 20 wells per group. OA, oleic acid. *P < 0.05, **P < 0.01, ***P < 0.001 for *STK25* siRNA *vs.* NTC siRNA.

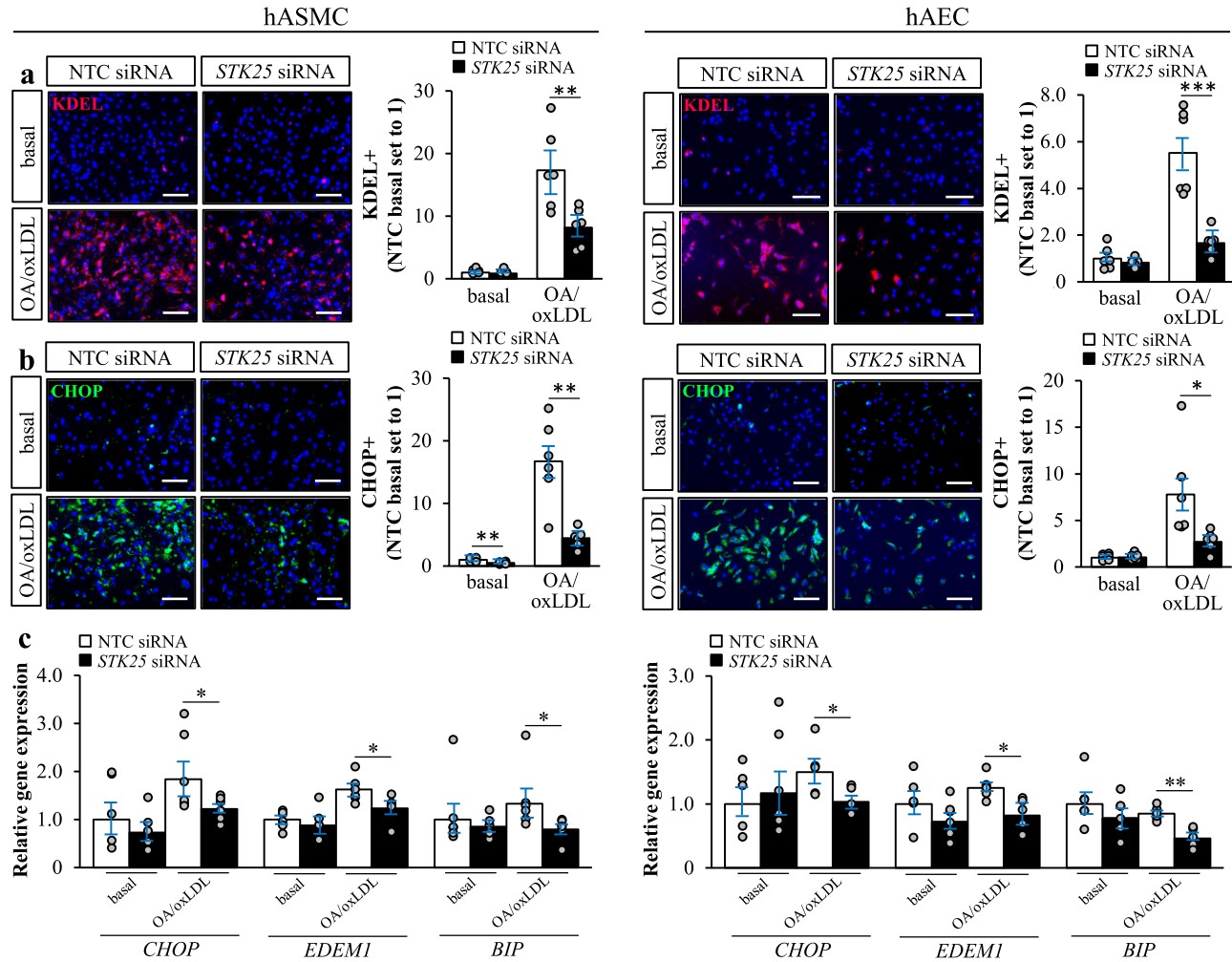

**Fig. 4 STK25 controls ER stress in human aortic endothelial and smooth muscle cells.** hASMCs and hAECs were transfected with *STK25* siRNA or NTC siRNA. **a, b** Representative images of cells processed for immunofluorescence with anti-KDEL (red) or anti-CHOP (green) antibodies; nuclei stained with DAPI (blue). Quantification of the staining. **c** Relative gene expression of selected ER stress markers. In (**a, b**), the scale bars represent 100. Data are mean ± SEM from 5 to 6 wells per group. OA, oleic acid. *P < 0.05, **P < 0.01, ***P < 0.001 for *STK25* siRNA vs. NTC siRNA.

Analysis of the sites of phosphorylation that were down-regulated in STK25-deficient hASMCs using the WebLogo software[37] showed that phosphorylation of serine residues was more frequent compared with threonine; however, no significant sequence similarity around the phosphoserine or -threonine residues could be identified (Fig. 7f). High variability at the phosphorylation sites is consistent with the experimental design, which does not allow for discriminate between the direct vs. indirect targets of STK25 activity.

Assessment of the known functions of the differentially represented proteins revealed an enrichment of targets linked to proliferation and migration, as well as calcium deposition and inflammation, in hASMCs transfected with *STK25* siRNA vs. NTC siRNA (Fig. 7c), which possibly contributed to the altered phenotypic switching, decreased calcification, and lower activation of inflammatory pathways observed in STK25-depleted cells. Surprisingly, AHSG (also known as fetuin-A), which is considered one of the main inhibitors of calcification[38], was lower in cells transfected with *STK25* siRNA. However, recent studies have suggested a dual action of AHSG, and this protein has also been shown to increase smooth muscle cell proliferation and collagen production, aggravating atherosclerotic plaque formation[39–41]. Of note, several members of the serpin protease inhibitor family were downregulated in STK25-deficient

hASMCs. Interestingly, out of these, SERPINA1, SERPINE1, and SERPINF1 are known to positively correlate with athero-sclerosis in humans[42–44].

Analysis of proteins containing differentially regulated sites of phosphorylation also revealed marked enrichment of targets known to regulate proliferation and migration, potentially contributing to the attenuated phenotypic transition detected in STK25-depleted hASMCs (Fig. 7e). We also found altered levels of phosphoproteins involved in the regulation of ER-associated protein degradation (ERAD), oxidative stress, and fibrosis in hASMCs transfected with *STK25* siRNA vs. NTC siRNA (Fig. 7e).

Enrichment analysis performed by integrating the proteomic data obtained from STK25-deficient hASMCs with the genome-scale metabolic model Human1[45] revealed changes in cellular dynamics both in terms of subsystems and reporter metabolite sets, which could potentially be affected by STK25 depletion. This analysis demonstrated that STK25 knockdown is associated with activation of fatty acid metabolism and oxidation subsystems and, in specific, with upregulation of β-oxidation pathways (Supplementary Fig. 9a), which is well-aligned with the enhanced β-oxidation rate detected in STK25-deficient hASMCs using palmitic acid as substrate (Fig. 1e). In addition, the enrichment analysis suggested co-upregulation of amino acid metabolism pathways by silencing STK25, implying close relations to fatty

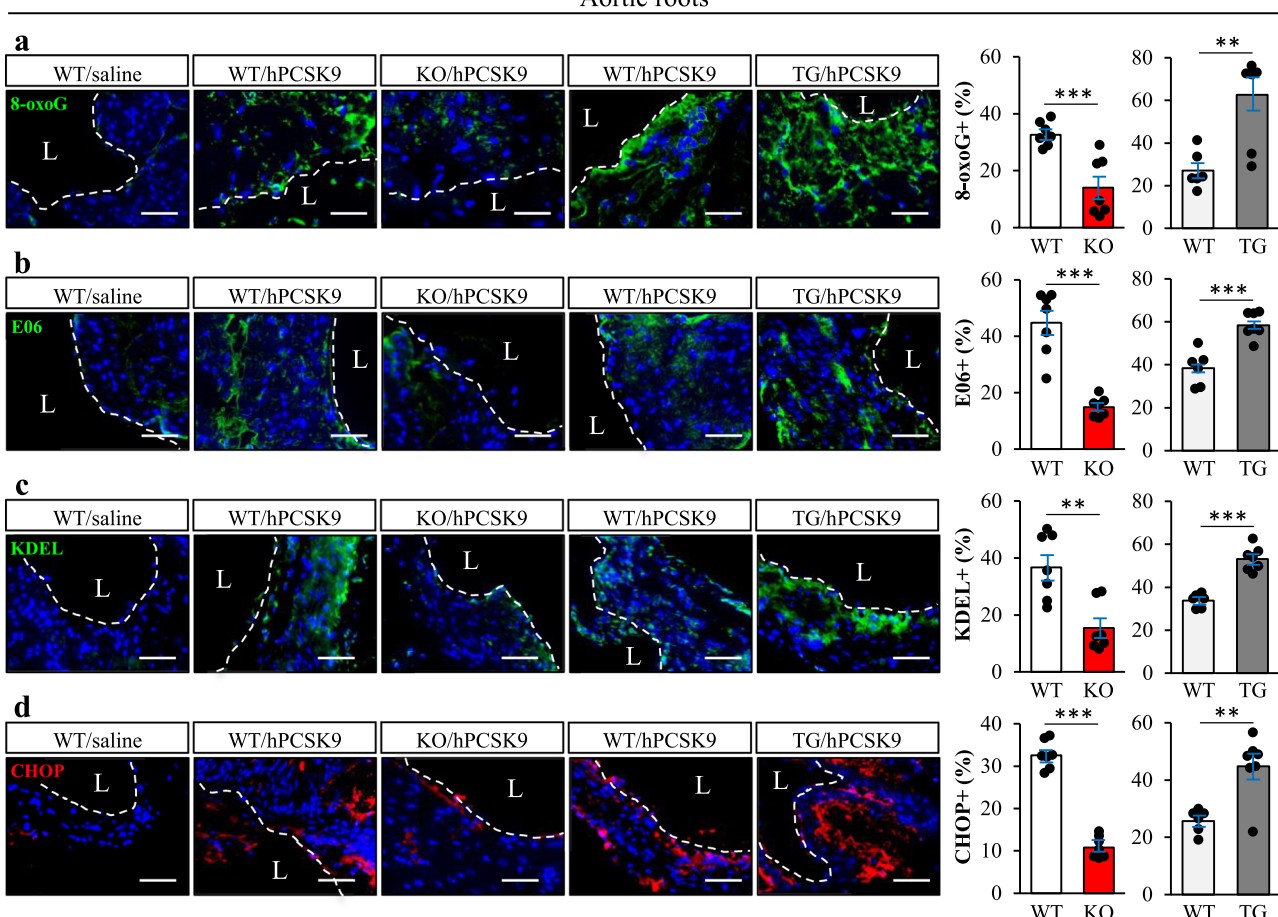

**Fig. 5 STK25 modulates aortic oxidative and ER stress in vivo.** Atherosclerotic *Stk25* knockout and transgenic mice, as well as wild-type littermates, were generated by gene transfer of gain-of-function mutant of PCSK9 combined with an atherogenic western-type diet. Saline-injected wild-type mice fed the western-type diet were included as controls. **a–d** Representative images of aortic sinus sections processed for immunofluorescence with anti-8-oxoG (green), anti-E06 (green), anti-KDEL (green), or anti-CHOP (red) antibodies; nuclei stained with DAPI (blue). Dashed line and L indicate lumen. The scale bars represent 50 μm. Negative controls by excluding primary antibody or by substituting a primary antibody by an equivalent concentration of the corresponding Ig isotype (Supplementary Fig. 11a, b) did not detect any positive signal, confirming the specificity of the immunostaining. Quantification of the staining. Data are mean ± SEM from 6-7 mice per group. L, lumen. **P < 0.01, ***P < 0.001 for knockout vs. wild-type mice and transgenic vs. wild-type mice.

acid metabolism (Supplementary Fig. 9a). The elevations of these subsystems were coherently reflected by the enrichment in sets of reporter metabolites representing various forms of CoA, such as acetyl-CoA, which is the hub metabolite of lipid metabolism (Supplementary Fig. 9b).

## Discussion

This study describes the cell-autonomous effects of STE20-type kinase STK25 in two of the main cell types of vasculature, human endothelial and smooth muscle cells. We found that silencing of STK25 in both cell types by an siRNA approach resulted in coordinated changes in several atheroprotective processes, hindering the transition of the cells to a dysfunctional atheroprone state (Fig. 8). The study provides mechanistic insights into our previous in vivo investigations in a mouse model of atherosclerosis, where genetic ablation of STK25 efficiently protected against the initiation and aggravation of aortic lesion formation[20].

In both endothelial and smooth muscle cells, the silencing of STK25 protected against ectopic lipid accumulation by decreasing fatty acid uptake and TAG synthesis as well as increasing the rate of lipolysis and β-oxidation. This is interesting in the light of recent evidence demonstrating that lipid storage in vascular cells

triggers inflammatory signaling as well as oxidative and ER stress, together fueling atherogenesis[21,22]. Indeed, in parallel with lower lipid deposition, we found reduced activation of inflammatory pathways in STK25-deficient cells, as evidenced by suppressed nuclear translocation of NFκB, less secretion of cytokines and chemokines, and attenuated monocyte adhesion. Furthermore, knockdown of STK25 in endothelial and smooth muscle cells resulted in suppression of several classical markers of oxidative and ER stress including DHE, 8-oxoG, E06, KDEL, and CHOP. Thus, the depletion of STK25 protects the main vascular cell types against lipotoxic damage by inducing a shift in the metabolic balance from lipid anabolism to lipid catabolism.

During plaque progression, the aortic cells are known to undergo osteogenic transdifferentiation leading to increased calcium deposition[31,46]. Interestingly, we found that depletion of STK25 in smooth muscle cells markedly suppressed calcification. Furthermore, we detected lower rates of proliferation and migration in STK25-deficient smooth muscle cells, indicating that STK25 inactivation hinders the shift from a contractile to a synthetic phenotype[27,28]. Global proteomic analysis revealed that these changes were accompanied by differential representation of several key markers of proliferation and migration as well as

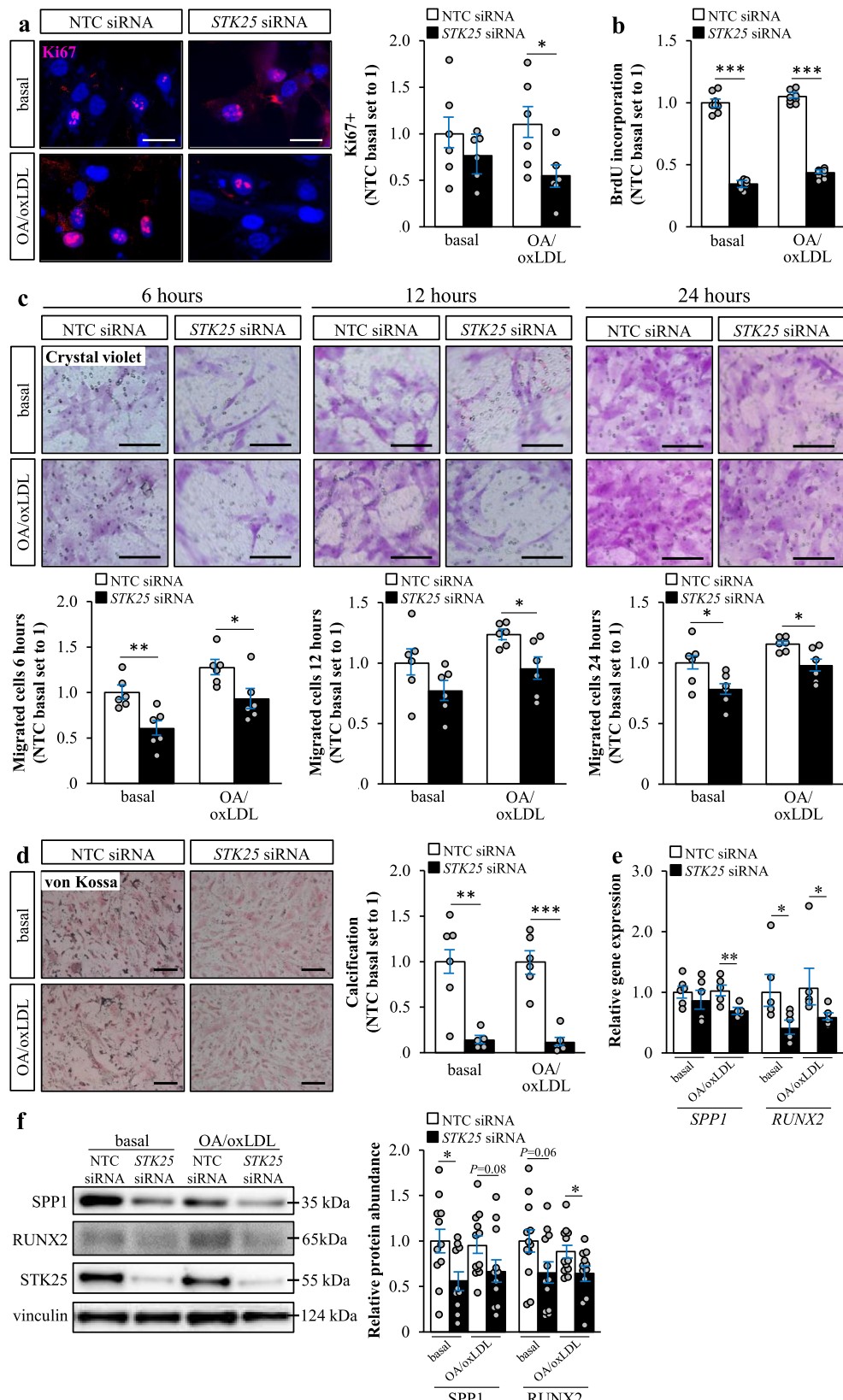

calcification in smooth muscle cells transfected with *STK25 vs.* control siRNA. However, the exact mechanisms by which STK25 regulates transdifferentiation and phenotypic switching in smooth muscle cells still remain elusive.

In addition to reduced lipid accumulation in STK25-deficient endothelial and smooth muscle cells reported in this study, we have

previously shown that antagonizing STK25 signaling also mitigates ectopic fat storage in liver, skeletal muscle, and kidney[9,10,12,14,16,18]. Of note, in spite of this similar functional readout, the mode of action of STK25 in the regulation of cellular lipid deposition appears to be different in different tissues. To this end, we found that the abundance of acetyl-CoA carboxylase (ACC), which

**Fig. 6 Silencing of STK25 decreases phenotypic switching in human aortic smooth muscle cells.** hASMCs were transfected with *STK25* siRNA or NTC siRNA. **a** Representative images of cells processed for immunofluorescence with anti-Ki67 (pink) antibody; nuclei stained with DAPI (blue). Quantification of the staining. **b** Proliferation rate was assessed by measuring the DNA synthesis using a Biotrack cell proliferation ELISA. **c** Representative images of migrated cells stained with crystal violet after 6, 12, and 24 h. Quantification of the staining. **d** Representative images of calcium deposition. Quantification of the staining. **e**, **f** Relative mRNA **e** and protein **f** abundance of selected calcification markers. Protein levels were analyzed by densitometry; representative Western blots are shown with vinculin used as a loading control. In (**a**, **c**, **d**), the scale bars represent 25 and 50 µm, respectively. Data in (**a**–**e**) are mean ± SEM from 5 to 8 wells per group; data in (**f**) are mean ± SEM from 11 to 12 wells per group. OA, oleic acid. *$P < 0.05$, **$P < 0.01$, ***$P < 0.001$ for *STK25* siRNA *vs.* NTC siRNA.

suppresses lipid oxidation and stimulates lipid synthesis[47], is decreased in liver cells where STK25 is knocked down[10,12,16,20] but not in STK25-deficient endothelial or smooth muscle cells (Supplementary Fig. 10). Interestingly, in contrast to the similar impact on intracellular fat storage in several tissues, multiple functional effects of STK25 vary across diverse cell types. As an example, we here found that silencing of STK25 robustly reduced the proliferation rate and calcification in smooth muscle but not in endothelial cells. The mechanisms behind these cell-type dependent roles of STK25 remain unknown; however, it is likely that specificity is achieved via the interactions of ubiquitously expressed STK25 with different tissue-specific partner proteins.

The development of novel therapeutic approaches for treating atherosclerosis, and reducing major clinical consequences such as myocardial infarction or stroke, will be dependent on a rigorous understanding of the biology of each of the major cell types that contribute to the pathogenesis of atherosclerotic lesions. Thus, a critical challenge for future studies will be to comprehensively decipher the molecular mechanisms that regulate phenotypic transitions of endothelial and smooth muscle cells from an atheroprotective to atheroprone state, and to determine how these might be effectively manipulated to reduce plaque burden and increase plaque stability. Here we provide, to our knowledge, the first evidence for an essential cell-autonomous role of protein kinase STK25 in atherosclerosis susceptibility and show that antagonizing STK25 signaling in human aortic endothelial and smooth muscle cells is atheroprotective. Future studies are needed to address the potential therapeutic relevance of pharmacological STK25 antagonism as a strategy to dampen or abrogate the development of atherosclerosis.

## Methods

**Cell culture and RNA interference**. Primary hASMCs (SC6110; 3H Biomedical, Uppsala, Sweden) were maintained in complete Smooth Muscle Cell Medium (3H Biomedical). Primary hAECs (304-05 A; Cell Applications Inc., Sigma-Aldrich, St. Louis, MO) were maintained in Medium 200 (Gibco, Waltham, MA) supplemented with Low Serum Growth Supplement (Gibco). Primary mASMCs (C57-6080; Cell Biologics, Chicago, IL) were maintained in complete Smooth Muscle Cell Medium (Cell Biologics). Primary mAECs (C57-6052; Cell Biologics) were maintained in complete Mouse Endothelial Cell Medium (Cell Biologics). THP-1 monocytes (TIB-202; American Type Culture Collection, Manassas, VA) were maintained in RPMI-1640 Medium (Gibco) supplemented with 10% (vol/vol) FBS (Gibco) and 0.05 mmol/l 2-mercaptoethanol (Sigma-Aldrich). Primary cells were used for experiments between passage 2 and 10. Cells were demonstrated to be free of mycoplasma infection by the MycoAlert Mycoplasma Detection Kit (LT07-218; Lonza, Basel, Switzerland).

Primary hASMCs and hAECs were transfected with human *STK25* siRNA (s20570; Ambion, Austin, TX) and primary mASMCs and mAECs were transfected with mouse *Stk25* siRNA (s81846; Ambion) using Lipofectamine RNAiMax (Thermo Fisher Scientific, Waltham, MA). As control, cells were transfected with scrambled siRNA (SIC001; Sigma-Aldrich). In all experiments, cells were exposed to 100 µg/ml oxLDL (Thermo Fisher Scientific) and/or 100 µmol/l oleic acid (Sigma-Aldrich) for 24 h, to mimic the conditions in high-risk patients.

**Mice**. Whole-body *Stk25* transgenic and knockout mice were generated and genotyped as previously described[8,48]. Mice were weaned at 3 weeks of age and housed 3 to 5 per cage in a temperature-controlled (21 °C) facility with a 12-h light-dark cycle and free access to chow and water. To induce atherosclerosis, *Stk25* transgenic and knockout mice, and their corresponding wild-type littermates, were given a tail vein injection of an adeno associated virus (rAAV8) carrying the gain-of-function mutant (Asp374 to Tyr) of human PCSK9 (hPCSK9$^{D374Y}$; $2 \times 10^{11}$ genome copies; kindly provided by Dr. Juan A. Bernal, Madrid, Spain)[49], and fed a western-type diet (21.2% fat and 0.2% cholesterol, TD.88137; Harlan Teklad, Madison, WI) for 12 weeks as previously described[20]. For comparison, a separate cohort of wild-type mice were given a tail vein injection of saline followed by a western-type diet feeding for 12 weeks. The Asp374 to Tyr mutation enhances the affinity of PCSK9 for LDLR by ≥10-fold, which has been shown to induce post-translational downregulation of hepatic LDLR, causing hypercholesterolemia and subsequent atherosclerosis in mice when combined with an atherogenic western-type diet[49–53]. In all experiments, *Stk25* transgenic and knockout mice were compared with their corresponding wild-type littermates since the genetic background of these lines differs (C57BL6/N for *Stk25* transgenic mice and C57BL6/J for *Stk25*$^{-/-}$ mice). Euthanasia was performed under full anesthesia with isoflurane 1–3% by removal of the heart in accordance with institutional and national regulations. No other procedures requiring anesthetic agents were used. The mice received human care according to National Institutes of Health (NIH) recommendations outlined in the Guide for the Care and Use of Laboratory Animals. All experiments were performed after prior approval from the Research Animal Ethics Committee in Gothenburg, Sweden.

**Assessment of lipid metabolism and lipotoxicity**. Cells were stained with Bodipy 493/503 (Invitrogen, Carlsbad, CA) for neutral lipids, or DHE (Life Technologies, Grand Island, NY) for superoxide radicals as previously described[19]. In parallel, cells and tissues were processed for immunofluorescence with anti-STK25, anti-NFκB, anti-8-oxoG, anti-E06, anti-KDEL, anti-CHOP, or anti-Ki67 antibodies (see Supplementary Table 1 for antibody information). The labeled area was quantified in 6–10 randomly selected microscopic fields (×20) using the ImageJ software (1.47 v; NIH, Bethesda, MD).

To measure β-oxidation, cells were incubated in the presence of (9,10-$^3$H[N]) palmitic acid (PerkinElmer, Waltham, MA), and [$^3$H]-labeled water was quantified as the product of free fatty acid oxidation[14]. TAG synthesis was measured as previously described[14]. Fatty acid uptake was quantified using the Quencher-Based Technology (QBT) Fatty Acid Uptake Assay Kit (R8132; Molecular Devices, San Jose, CA) according to the manufacturer's recommendations. The TAG hydrolase activity was determined in total cell lysates using [$^3$H]triolein (PerkinElmer) as the substrate in an assay buffer containing 0.25 mmol/l triolein, 0.8 mmol/l phosphatidylcholine, 20 mmol/l Tris, 150 mmol/l NaCl, and 1 mmol/l EDTA, pH 8.0 (Sigma-Aldrich).

**Biochemical assays**. Primary hASMCs and hAECs were treated with 1 µg/ml of lipopolysaccharide (LPS; Thermo Fisher Scientific) for 24 hours. The supernatants of conditioned media were collected, and the concentration of MCP-1, IL-1β, IL-8, and TNFα was measured using the Human MCP-1, Human IL-1β, Human IL-8, and Human TNFα ELISA Kits (ab100586, ab46052, ab229402, ab181421; all from Abcam, Cambridge, UK), respectively. Additionally, the cellular ROS and NO content was measured using the DCFDA/H2DCFDA Cellular ROS Assay Kit and NO Assay (Orange) Kit (ab113851, ab219932; both from Abcam), respectively[54,55].

Primary hASMCs and hAECs were treated with human TGF-β1 (2.5 ng/ml; R&D Systems, Minneapolis, MN) for 24 h. Cells were collected for analysis of fibrotic gene expression (see below) and the concentration of collagen in the media was measured using the Soluble Collagen Assay Kit (K532-100-BV; BioVision, Mountain View, CA).

Phase extraction of metabolites from hASMCs and hAECs was performed as previously described[56]. The AbsoluteIDQ p180 Kit (20714-130; Biocrates Life Sciences, Innsbruck, Austria) was used to determine 186 metabolites. The samples were measured on mass spectrometer QTRAP 4500 (Sciex, Framingham, MA), in combination with a high-performance LC (Agilent Technologies, Waldbronn, Germany). The concentrations and ratios of the metabolites were calculated automatically by the MetIDQ software (Biocrates Life Sciences).

**Monocyte adhesion assay**. THP-1 cells were labeled with 2.5 µmol/l calcein acetoxymethyl (Sigma-Aldrich) for 30 min. The labeled cells were washed with PBS 3 times to remove the residual calcein acetoxymethyl. Calcein acetoxymethyl-labeled THP-1 cells ($1 \times 10^5$) were added to a confluent monolayer of hASMCs and hAECs for 1 h, and then washed with PBS 3 times.

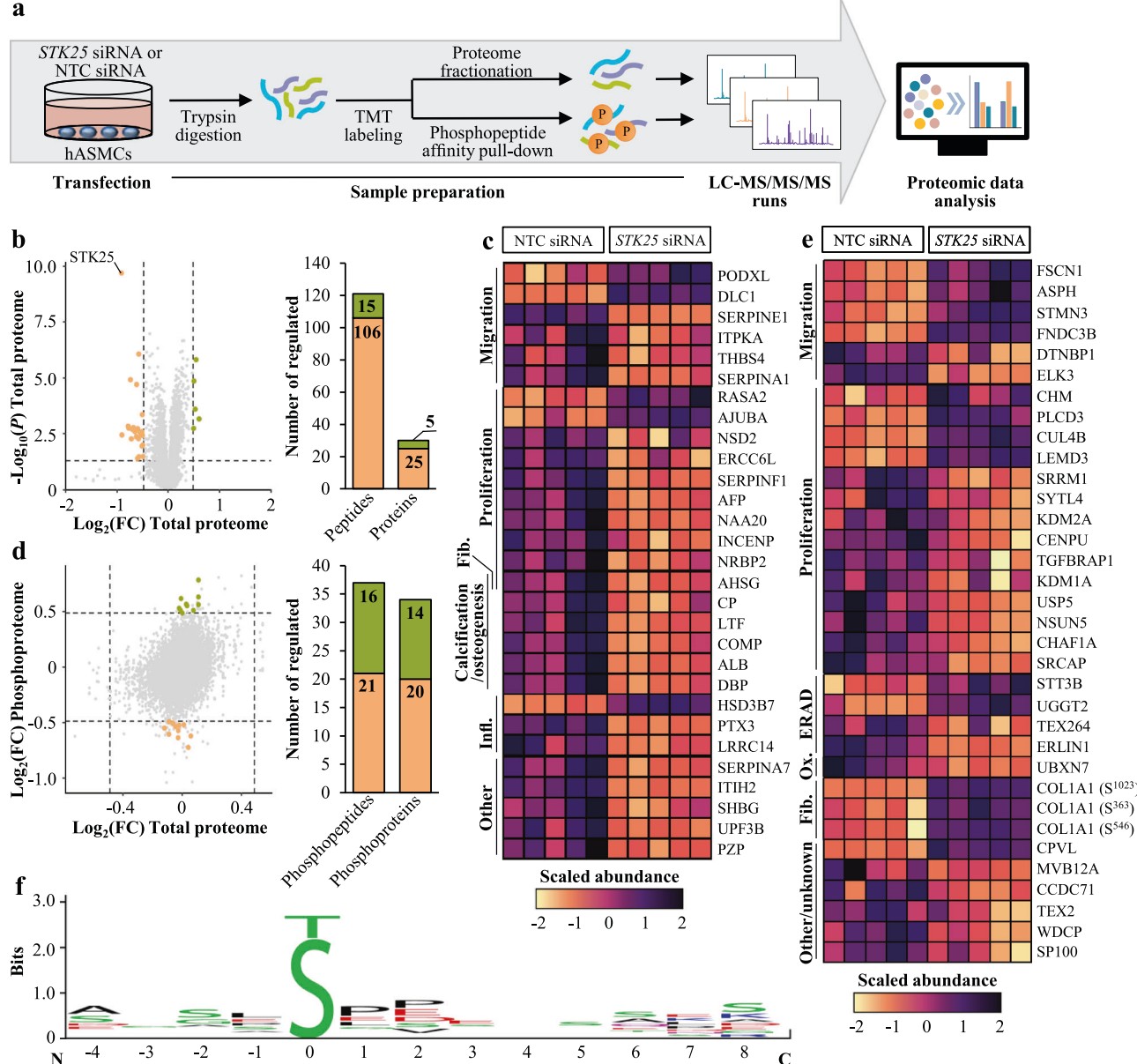

**Fig. 7 Global proteomics and phosphoproteomic analysis in human aortic smooth muscle cells.** hASMCs were transfected with *STK25* siRNA or NTC siRNA. **a** Schematic presentation of the experimental design. **b** Volcano plot of the log-transformed fold change and significance of differentially enriched proteins. A ratio of 1.4-fold (vertical dashed lines) and a *P* value of 0.05 (horizontal dashed line) serve as the threshold for differential expression. Stacked bar plot showing the total number of differentially expressed unique peptides and proteins. **c** Heat map of the scaled abundance of significantly changed proteins. The functions are indicated to the left. **d** Scatter plot of the log-transformed fold change of total protein abundance and the log-transformed fold change of phosphoproteins. Stacked bar plot showing the total number of differentially expressed unique phosphopeptides and phosphoproteins that were not driven by underlying changes in the abundance of corresponding proteins. A ratio of 1.4-fold (vertical and horizontal dashed lines) and a *P* value of 0.05 serve as the threshold for differential expression. **e** Heat map of the scaled abundance of significantly changed phosphoproteins. The functions are indicated to the left. **f** Consensus sequences were extracted from the peptides with downregulated phosphorylation events in STK25-depleted hASMCs using the WebLogo application[37]. The residue position in relation to the phosphorylation site is shown on the X-axis and the information content is shown on the Y-axis, where the height of each position is representing the certainty level of possibly presented residues in that position. For empty positions, there were not enough information to determine its composition. The colors of the amino acids correspond to their chemical properties; polar amino acids (G, S, T, Y, C, Q, and N) are shown in green, basic amino acids (K, R, and H) are shown in blue, acidic amino acids (D and E) are shown in red, and hydrophobic amino acids (A, V, L, I, P, W, F, and M) are shown in black. FC, fold change; Fib., fibrosis; Infl., inflammation; Ox., oxidative stress.

Adherent monocytes were quantified in 6 randomly selected microscopic fields (×20) using the ImageJ software.

**Cell proliferation**. Proliferation was measured using the BrdU-Labeling Assay Kit (Amersham Cell Proliferation Biotrak ELISA System, version 2, RPN250; GE Healthcare, Chicago, IL) according to the manufacturer's recommendations. Briefly, cells were incubated with BrdU-labeling solution for 2 h. After removing

the labeling solution, the cells were fixed, blocked and incubated with peroxidase labeled anti-BrdU solution, followed by incubation with TMB substrate. The reaction was stopped with 1 mol/l sulphuric acid and the absorbance was measured using the SpectraMax iD3 Multi-Mode Microplate Reader (Molecular Devices).

**Migration**. Primary hASMCs and hAECs were added to transwell chambers with 8 μm pore size (Nunc Polycarbonate Cell Culture Inserts in Multi-Well Plates;

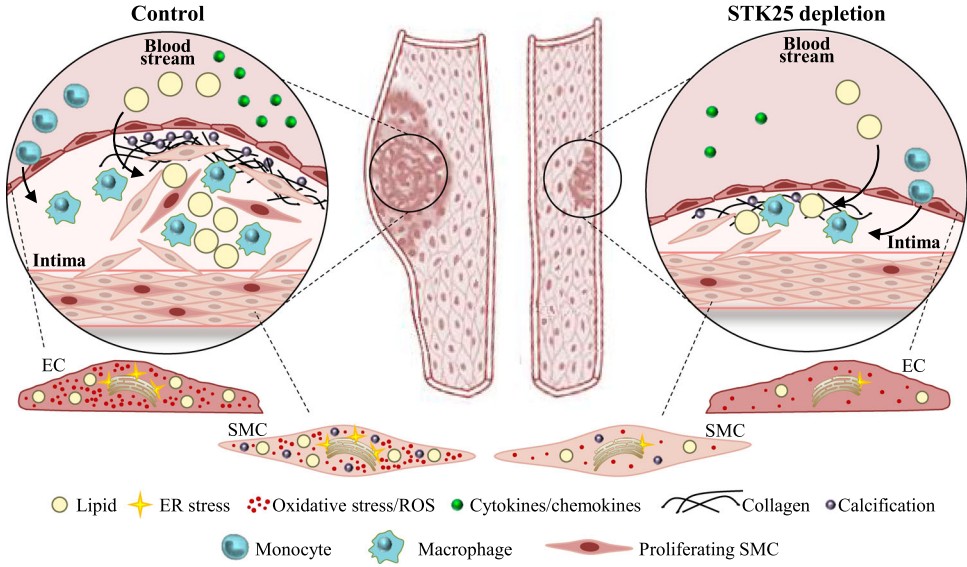

**Control**　　　　**STK25 depletion**

Blood stream　　　Intima　　　EC　　　SMC

○ Lipid　　✦ ER stress　　⋰Oxidative stress/ROS　　● Cytokines/chemokines　　⟋⟍Collagen　　●Calcification

◉ Monocyte　　◉ Macrophage　　▬ Proliferating SMC

**Fig. 8 Schematic illustration of the impact of STK25 in atherosclerosis susceptibility.** Silencing of STK25 in human aortic endothelial and smooth muscle cells protects against the atheroprone changes that lead to atherosclerosis development and progression.

Thermo Fisher Scientific). After 6, 12, and 24 h, cells on the upper surface were removed using cotton swabs, and the cell on the bottom side of the membranes were fixed and stained with 0.1% crystal violet (Sigma-Aldrich). Migrated cells were counted in 6 randomly selected microscopic fields (×10) using the ImageJ software.

**Calcification.** Calcification was measured using von Kossa staining. Briefly, primary hASMCs and hAECs were incubated with a solution of 1% silver nitrate (Sigma-Aldrich) and exposed to light until turning brown; staining was done with 5% sodium thiosulfate and nuclear fast red (Sigma-Aldrich)[57]. Calcification was analyzed in 6 randomly selected microscopic field (×10) using the ImageJ software.

**qRT-PCR and Western Blot.** The methods used for the isolation of RNA, the synthesis of cDNA, and the qRT-PCR assay have been described previously[58]. Nuclear and cytoplasmic extracts were prepared using the NE-PER Nuclear and Cytoplasmic Extraction Reagents (Thermo Fisher Scientific) according to the manufacturer's recommendations. Western blotting was carried out as previously described[8] (see Supplementary Table 1 for antibody information).

**LC-MS analysis of smooth muscle cells.** The cells were lysed in 2% sodium dodecyl sulfate, 50 mmol/l triethylammonium bicarbonate (TEAB), and X1 Pierce Phosphatase Inhibitor (Thermo Fisher Scientific). A representative reference pool containing equal amounts from 10 of the samples was made.

Aliquots containing 200 µg of total protein were incubated at 37 °C for 60 min in the lysis buffer with DL-dithiothreitol (DDT) at 100 mmol/l final concentration. The reduced samples were processed using the modified filter-aided sample preparation method[59]. Briefly, samples were reduced with 100 mmol/l DDT at 37 °C for 60 min, transferred to 30 kDa MWCO Pall Nanosep Centrifugation Filters (Sigma-Aldrich), washed repeatedly with 8 mol/l urea and once with digestion buffer [0.5% sodium deoxycholate (SDC) in 50 mmol/l TEAB] prior to alkylation with 10 mmol/l methyl methanethiosulfonate in digestion buffer for 30 min. Digestion was performed in digestion buffer by addition of Pierce MS Grade Trypsin (Thermo Fisher Scientific) in an enzyme to protein ratio of 1:100 at 37 °C for 3 h. An additional portion of trypsin was added and incubated overnight. Peptides were collected by centrifugation.

Digested peptides were labeled using tandem mass tag (TMT) 11-plex isobaric mass tagging reagents (Thermo Fisher Scientific) according to the manufacturer's recommendations. Samples were combined into two TMT-sets, each including two groups and a reference. SDC was removed by acidification with 10% trifluoroacetic acid and peptides were desalted using Pierce Peptide Desalting Spin Columns (Thermo Fisher Scientific) according to the manufacturer's recommendations.

An aliquot corresponding to 200 µg was withdrawn for the total proteome analysis, and an aliquot of 1 mg was subjected to phosphopeptide enrichment using the High-Select Fe-NTA Enrichment Kit and another 1 mg aliquot was treated with the High-Select TiO₂ Phosphopeptide Enrichment Kit (A32992, A32993; both Thermo Fisher Scientific) according to the manufacturer's recommendations. The eluted phosphopeptide samples were pooled and pre-fractionated into 20 fractions, whereas the combined sets for total proteome were pre-fractionated into 40 fractions. Peptide separations were performed using basic reversed-phase chromatography with a Dionex Ultimate 3000 UPLC System (Thermo Fisher Scientific) on a reversed-phase XBridge BEH C18 Column (3.5 µm, 3.0 × 150 mm; Waters Corporation, Milford,

MA). Total proteome samples were fractionated using 10 mmol/l ammonium formate buffer at pH 10.00 as solvent A and 90% acetonitrile/10% 10 mmol/l ammonium formate at pH 10.00 as solvent B, via a linear gradient from 3% to 40% B over 18 min followed by an increase to 100% B over 5 min, whereas phosphopeptides were separated from 3% to 40% B over 12 min and an increase to 100% B over 4 min. The fractions of the phosphoproteomics samples were concatenated into 10 fractions while the whole proteomics fractions were concatenated into 20 fractions, dried and reconstituted in 3% acetonitrile/0.2% formic acid.

The fractions of the whole proteome samples were analyzed on an Orbitrap Fusion Tribrid Mass Spectrometer interfaced with Easy-nLC1200 LC System (Thermo Fisher Scientific). Peptides were trapped on an Acclaim Pepmap 100 C18 Trap Column (100 µm × 2 cm, particle size 5 µm; Thermo Fisher Scientific) and separated on an in-house packed analytical column (75 µm × 40 cm, particle size 3 µm, Reprosil-Pur C18; Dr. Maisch, Ammerbuch, Germany) using 0.2% formic acid as solvent A and 80% acetonitrile/0.2% formic acid as solvent B, via a linear gradient from 5% to 33% B over 75 min followed by an increase to 100% B for 5 min, and 100% B for 10 min at a flow of 300 nl/min. Precursor ion mass spectra were acquired at 120,000 resolution and MS/MS analysis was performed in a data-dependent multinotch mode where CID spectra of the most intense precursor ions were recorded in ion trap at normalized collision energy setting of 35% for 3 s ('top speed' setting). Precursors were isolated in the quadrupole with a 0.7 m/z isolation window, charge states 2 to 7 were selected for fragmentation, and dynamic exclusion was set to 60 s and 10 ppm. MS3 spectra for reporter ion quantitation were recorded at a resolution of 50000 with higher-energy collision dissociation (HCD) fragmentation at normalized collision energy of HCD set to 65% using the synchronous precursor selection. Analyses of the phosphoproteomics samples were carried out in a similar manner except that the most abundant doubly or multiply charged precursors from the MS1 scans were isolated with a dynamic exclusion within 10 ppm for 45 s. The isolated precursors were fragmented by HCD at 38% collision energy and the MS2 spectra were detected in the Orbitrap at 50000 resolution, with the fixed first m/z 100 and maximum injection time 150 ms.

Data analysis was performed using Proteome Discoverer version 2.4 (Thermo Fisher Scientific). Identification was performed using Mascot version 2.5.1 (Matrix Science, London, UK) as a search engine by matching against the *Homo sapiens* database of SwissProt. The precursor mass tolerance was set to 5 ppm and fragment mass tolerance to 0.6 Da. Tryptic peptides were accepted with zero missed cleavage, variable modifications of methionine oxidation and fixed cysteine alkylation, and TMT-label modifications of N-terminal and lysine were selected. For the phosphoproteomics samples, variable modification of serine, threonine, or tyrosine phosphorylation was added. The reference samples were used as denominator and for calculation of the ratios. Percolator was used for the validation of identified proteins.

Quantification was performed in Proteome Discoverer 2.4. TMT reporter ions were identified with 3 mmu mass tolerance in the MS2 HCD spectra for the phosphopeptide experiment or in the MS3 HCD spectra for the total proteome experiment, and the TMT reporter S/N values for each sample were normalized within Proteome Discoverer 2.4 on the total peptide amount. Only the unique identified peptides were taken into account for the protein quantification.

**Integrative analysis of proteomic data with human-GEM.** Subsystem (pathway) and reporter metabolite gene sets were retrieved from Human-GEM v1.10.0[60] and

integrated with proteomic data obtained from hASMCs for enrichment analysis using the GSAM package[61]. The approach used for estimating the significance of directional gene set enrichment has been described previously[62]. Gene sets were first scored using a reporter method[63], and the enrichment significance of each gene set was estimated by comparing the scores with 50000 randomly shuffled gene sets of equal size.

**Statistics and reproducibility.** Statistical significance between the groups was evaluated by using one-way ANOVA with a two-sample Student's *t* test for post hoc analysis. The Shapiro-Wilk's and the Levene's tests were applied to confirm the normality of distribution of residuals and the homogeneity of variances, respectively. Differences were considered statistically significant at $P < 0.05$. Samples sizes were predetermined based on statistical power calculations or convention in the field. The exact sample size is given in the legend of each figure. All statistical analyses were conducted using SPSS statistics (v24; IBM Corporation, Armonk, NY) or R (The R Foundation, Vienna, Austria).

**Reporting summary.** Further information on research design is available in the Nature Research Reporting Summary linked to this article.

## Data availability

The mass spectrometry proteomics data have been deposited to the ProteomeXchange Consortium via the PRIDE[64] partner repository (dataset identifier PXD031763). The source data for the graphs in the main figures are available in Supplementary Data 1. Uncropped Western blot images are provided in Supplementary Fig. 12. Any remaining information can be obtained from the corresponding author upon reasonable request.

## Code availability

The code and data for the enrichment analysis are available at https://bitbucket.org/scilifelab-lts/m_mahlapuu_2005/.

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

## Acknowledgements
The Proteomics Core Facility at Sahlgrenska Academy, University of Gothenburg, Sweden are grateful of Inga-Britt and Arne Lundbergs Forskningsstiftelse for the donation of the Orbitrap Fusion Tribrid MS instrument. This work was supported by grants from the Swedish Heart-Lung Foundation; the Swedish Research Council; the Swedish Cancer Society; the European Foundation for the Study of Diabetes; the West Sweden Avtal om Läkarutbildning och Forskning (ALF) Program, the Novo Nordisk Foundation; the Swedish Diabetes Foundation; the Å. Wiberg Foundation; the Magnus Bergvall Foundation; the Adlerbert Research Foundation; the I. Hultman Foundation; the F. Neubergh Foundation; the Prof. N. Svartz Foundation; the L. and J. Grönberg Foundation; the W. and M. Lundgren Foundation; and the I.-B. and A. Lundberg Research Foundation. J.R. and H.W. are financially supported by the Knut and Alice Wallenberg Foundation as part of the National Bioinformatics Infrastructure Sweden at SciLifeLab.

## Author contributions
E.C. designed the study and generated the bulk of the results, and wrote, reviewed, and edited the manuscript. S.K., M.C., Y.X., R.P., B.M.O., and J.V. contributed to the research data and reviewed and edited the manuscript. J.R. and H.W. performed the phospho-proteomic data analysis and reviewed and edited the manuscript. J.G., U.S., L.T.S., C.S., H.U.M., A.E., and I.G. provided advice and expertise, and reviewed and edited the manuscript. M.M. directed the project, designed the study, interpreted the data, and wrote, critically reviewed, and edited the manuscript. M.M. is the guarantor of this work and, as such, had full access to all the data in the study and takes responsibility for the integrity of the data and the accuracy of the data analysis.

## Funding

## Competing interests
The authors declare no competing interests.
