## [Peer Review File · Communications Biology]

Reviewers' comments:

Reviewer #1 (Remarks to the Author):

The Manuscript ID# COMMSBIO-21-2165-T, titled "Silencing of STE20-Type Kinase STK25 in Human Aortic Cells is Atheroprotective" by Cansby et al., have presented interesting results of targeting STK25 in reducing fatty acid induced atherogenicity and fibrosis. However, there are few minor issues to be addressed prior its publication.

1. This study utilized with Aortic smooth muscle cells (ASMCs) and aortic endothelial cells (AECs). Using the term "aortic cells" is broader so, specifying SMCs and ECs is appropriate throughout the manuscript
2. It was not clear why the authors have used rAAV8-hPCSK9D374Y virus? Is this for overexpression of STK25 in knockout mice? please explain in the methods
3. Name the assay/method and cite appropriate reference for [³H] labelled water in the cells and TAG secretion measurements.
4. Cite the appropriate reference for DCF-DA and NO assay.
5. Include the cat# of all antibodies, assay kits.
6. Please define the abbreviations as it appears first time in the text.
7. Name the stain "Von Kossa" for calcification in methods and cite the appropriate reference.
8. There are few typos to be fixed. Eg. "50 000" should be 50000? And "20 000" may be 20000?
9. "at collision energy of 65 using the synchronous" include the units of the energy. It may be volts?
10. It was not clear how the FFA uptake was assessed while treating the cells with oleic acid and Oxy LDL. How it was different from the results presented in fig. 1 D, K to FM. A clear method could be included. And the results could be discussed.
11. TAG hydrolase activity was increased in cells with STK25 knockdown. Was this phenotype observed in STK25 KO mouse? could be discussed.
12. Authors can comment on ... There is significant reduction of IL8 in SMCs with STK25 knockdown, but it was not the case in a ECs?

Reviewer #2 (Remarks to the Author):

The STE20-type kinase (STK25) was recently shown to promote ectopic lipid storage and associated cell injury in several metabolic organs. While the global deletion of STK20 was found to decrease atherosclerotic lesion formation and macrophage infiltration, the role of STK25 in vascular cell types is unclear. Cansby E et al. utilize various in vitro techniques to assess the impact of STK25 loss in both smooth muscle cells and endothelial cells, where antagonizing STK25 may facilitate an atheroprotective effect in vascular SMCs.

Limitations and Strengths:

This study aimed to explore the role of STK25 in vascular cell types. The study integrates several classic in vitro techniques together with a phospho-proteomic screen to delineate the mechanistic role of STK20 in SMCs. However, the lack of validation and follow up of the screen hits limits the impact of this study, as many of the findings were descriptive and arguably expected, given the recent in vivo observations (Cansby E et al, ATVB, 2018).

Major Comments:

1. Figure2A,F: The nuclear localization of NF- κ B is difficult to appreciate in these images. It would help to either include a nuclear envelope marker or show only the co-localized nuclear signal. Nuclear/cytoplasmic fractionation is another option to reinforce this idea. Also, why are the cells so subconfluent? The hSMCs show a single cell.
2. Figure5C: The difference in migration is difficult to see. Considering that loss of STK25 results in impaired SMC proliferation, how can one be sure that the decreased migration seen after 24hrs isn't due to the impaired SMC growth? Alternative migration assays on a shorter time scale would help dissect these two SMC functions.
3. Figure5: While the authors claim that STK25 maintains the SMC phenotype in a contractile

state, it would help to show protein expression of the prototypical SMC markers (e.g. SMMHC, SM22a, calponin, etc). What happens to myocardin expression with siSTK25?

4. Supplementary Figure6: How does the decrease in putrescine in SMCs relate to efferocytosis? (PMID: 32004476)

5. The authors present data to indicate cell-autonomous differences in vascular cell types (SMC verses EC) and would benefit from further discussion. How does a ubiquitously expressed kinase mediate unique effects depending on cell type?

6. It is unclear as to why a phospho-proteomic analyses were not performed for ECs in parallel. The manuscript is clearly laid out as a comparison of vascular cell types to better understand STK25 function. The inclusion of an endothelial screen would have greatly enhanced the paper as it would present plausible reasons to address the previous question. It is possible, however, that finances were a factor in study design. If indeed the case, then a follow up of the current SMC screen should be included to strengthen the findings of an overall descriptive paper.

Minor Comments:

1. The manuscript refers to Figure 4A prior to mention of Figure 3H.

Reviewer #3 (Remarks to the Author):

The manuscript entitled "Silencing of STE20-Type Kinase STK25 in human aortic cells is atheroprotective" by Cansby et al. demonstrates an atheroprotective role of STK25 deletion in human aortic SMC and EC primary cultures. Specifically, the authors show that STK25 deletion in HASMC and HAEC reduces lipid deposition accompanied with reduced inflammatory and fibrotic markers as well as lowered oxidative and ER stress. This is a well-written and well-packaged manuscript pertinent to pathogenesis of atherosclerosis. Earlier work by the group demonstrated an atheroprotective outcome of STK25 deletion in vivo; in the current manuscript, the authors extended those findings to two key vascular cell types (VSMC, EC) to delineate cell-specific regulatory role of STK25 in atherosclerosis. While the manuscript reveals interesting results, novelty is somewhat limited in the light of their earlier publication (ATVB 2018). However, a few additional experiments (noted below) would enhance the strength of this manuscript. Nevertheless, this is an interesting paper, well-articulated with excellent presentation of data.

Specific concerns

1. In vitro studies presented have solely utilized siRNA gene silencing in HASMC and HAEC. Manuscript would strengthen if findings can be confirmed in murine aortic SMC and EC from STK25 KO mice for cross-species validation. Such data would also provide direct evidence for a cell-autonomous mechanism underlying the atheroprotective effect of STK25 deletion reported by the authors earlier (ATVB 2018). I suggest that you repeat a few experiments (eg. Figs 1, 2, 5A, B) in murine cells from the STK25 KO mice.

2. How does STK25 deletion affect SMC proliferation in vivo? Cells in culture are known to behave differently from in vivo. Additional experiments using reporter mice combined with STK25 KO in vivo would greatly strengthen findings and confirm regulatory role of STK25 on VSMC phenotypic switching in vivo.

3. In Fig 5, how does STK25 affect Runx2, Spp1 at protein level. How about SMC contractile markers (Myh11, LMOD-1). what is the molecular mechanism by which STK25 regulates VSMC phenotypic switching? Additional experiments are required to delineate this aspect.

4. Fig. 3H, 3I, 4D, 4E: mark lumen side in images. Also provide corresponding H&E and specificity controls for IHC images.

Dear Dr. Ngan F. Huang,

We would like to thank the Editors and Reviewers for their constructive criticism, which we believe has helped us to improve the quality of our manuscript (MS). We are pleased to have the opportunity to submit a revised version of our MS entitled “*Silencing of STE20-Type Kinase STK25 in Human Aortic Cells is Atheroprotective*” (COMMSBIO-21-2165-T) and we hope that it is now satisfactory for publication. We have revised the MS according to the suggestions of the Reviewers, and below we have provided a point-by-point response to all comments. All changes and new material in the MS are highlighted in colour and by underlining.

Reviewer(s)’ Remarks to the Author:

Reviewer #1: The Manuscript ID# COMMSBIO-21-2165-T, titled “Silencing of STE20-Type Kinase STK25 in Human Aortic Cells is Atheroprotective” by Cansby et al., have presented interesting results of targeting STK25 in reducing fatty acid induced atherogenicity and fibrosis. However, there are few minor issues to be addressed prior it its publication.

Response: We appreciate this Reviewer’s encouraging overall comment. All specific criticisms and suggestions that we reply to step-by-step further on helped us to improve our MS.

1. This study utilized with Aortic smooth muscle cells (ASMCs) and aortic endothelial cells (AECs). Using the term “aortic cells” is broader so, specifying SMCs and ECs is appropriate throughout the manuscript

Response: We have now specified throughout the MS that we have used aortic smooth muscle cells and aortic endothelial cells as requested by the Reviewer.

2. It was not clear why the authors have used rAAV8-hPCSK9D374Y virus? Is this for overexpression of STK25 in knockout mice? please explain in the methods

Response: We have now explained the use of rAAV8-hPCSK9^{D374Y} virus in the Materials and Methods section of the revised MS as follows (page 5): “To induce atherosclerosis, Stk25 transgenic and knockout mice, and their corresponding wild-type littermates, were given a tail vein injection of an adeno associated virus (rAAV8) carrying the gain-of-function mutant (Asp374 to Tyr) of human PCSK9 (hPCSK9^{D374Y}; 2x10¹¹ genome copies; kindly provided by Dr. Juan A. Bernal, Madrid, Spain),²² and fed a western-type diet (21.2% fat and 0.2% cholesterol, TD.88137; Harlan Teklad, Madison, WI) for 12 weeks as previously described.²⁰ For comparison, a separate cohort of wild-type mice were given a tail vein injection of saline followed by a western-type diet feeding for 12 weeks. The Asp374 to Tyr mutation enhances the affinity of PCSK9 for LDLR by ≥10-fold, which has been shown to induce post-translational downregulation of hepatic LDLR, causing hypercholesterolemia and subsequent atherosclerosis in mice when combined with an atherogenic western-type diet.²²⁻²⁶”

3. Name the assay/method and cite appropriate refence for [3H] labelled water in the cells and TAG secretion measurements.

Response: We have now cited appropriate references for the methods used to measure β-oxidation and TAG secretion as requested by the Reviewer (Materials and Methods section at page 6).

4. Cite the appropriate reference for DCF-DA and NO assay.

Response: We have now cited appropriate references for DCF-DA and NO assays (Materials and Methods section at page 6). We apologize that this information was missing and we thank the Reviewer for pointing this out.

5. Include the cat# of all antibodies, assay kits.

Response: The catalogue numbers of assay kits have now been added to the Materials and Methods section of the revised MS (pages 4-9). The catalogue numbers of all antibodies are listed in Supplementary Table 1.

6. Please define the abbreviations as it appears first time in the text.

Response: We have now defined each abbreviation as it appears first time in the text. We apologize for these errors.

7. Name the stain “Von Kossa” for calcification in methods and cite the appropriate reference.

Response: We have now added the name and the appropriate reference for this staining (Materials and Methods section at page 8).

8. There are few typos to be fixed. Eg. “50 000” should be 50000? And “20 000” may be 20000?

Response: We have now corrected these numbers (Materials and Methods section at pages 9-10). We thank the Reviewer for pointing out these errors.

9. “at collision energy of 65 using the synchronous” include the units of the energy. It may be volts?

Response: We thank the Reviewer for this important comment. This sentence has now been corrected as follows (Materials and Methods section at page 10): “MS3 spectra for reporter ion quantitation were recorded at a resolution of 50000 with higher-energy collision dissociation (HCD) fragmentation at normalized collision energy of HCD set to 65% using the synchronous precursor selection.”

10. It was not clear how the FFA uptake was assessed while treating the cells with oleic acid and Oxy LDL. How it was different from the results presented in fig. 1 D, K to FM. A clear method could be included. And the results could be discussed.

Response: Lipid accumulation assessed by Bodipy staining was studied under basal culture conditions as well as after treatment of the cells with oleic acid only or with both oleic acid and oxLDL (Fig. 1D, K). In contrast, FFA uptake was measured at basal conditions and after treating the cells with both oleic acid and oxLDL (Fig. 1F, M). We have now clarified the experimental setup used, and expanded the discussion of the results, as requested by the Reviewer (Results section at pages 11-12).

11. TAG hydrolase activity was increased in cells with STK25 knockdown. Was this phenotype observed in STK25 KO mouse? could be discussed.

Response: We agree with the Reviewer that it would be very valuable to compare TAG hydrolase activity in aortic endothelial and smooth muscle cells collected from Stk25 knockout mice versus wild-type littermates. Unfortunately, no cohorts of Stk25^{-/-} and wild-type mice for harvesting these samples are currently available. Thus, performing this investigation would require about 12-18 months of additional studies (including breeding program and high-fat diet feeding) and we respectfully submit that this is beyond the scope of this 3 months revision.

12. Authors can comment on ... There is significant reduction of IL8 in SMCs with STK25 knockdown, but it was not the case in a ECs?

Response: We thank the Reviewer for this comment. Indeed, the levels of IL-8 were significantly decreased by the depletion of STK25 only in hASMCs, although there was a similar tendency in hAECs. We have now emphasized this difference in the Results section of the revised MS (page 12).

Reviewer #2: The STE20-type kinase (STK25) was recently shown to promote ectopic lipid storage and associated cell injury in several metabolic organs. While the global deletion of STK20 was found to decrease atherosclerotic lesion formation and macrophage infiltration, the role of STK25 in vascular cell types is unclear. Cansby E et al. utilize various in vitro techniques to assess the impact of STK25 loss in both smooth muscle cells and endothelial cells, where antagonizing STK25 may facilitate an atheroprotective effect in vascular SMCs.

Limitations and Strengths:

This study aimed to explore the role of STK25 in vascular cell types. The study integrates several classic in vitro techniques together with a phospho-proteomic screen to delineate the mechanistic role of STK20 in SMCs. However, the lack of validation and follow up of the screen hits limits the impact of this study, as many of the findings were descriptive and arguably expected, given the recent in vivo observations (Cansby E et al, ATVB, 2018).

Response: We would like to thank the Reviewer for very valuable comments that helped us to improve our MS. Below we have provided a point-by-point response to all specific criticisms and suggestions.

1. Figure2A,F: The nuclear localization of NF- κ B is difficult to appreciate in these images. It would help to either include a nuclear envelope marker or show only the co-localized nuclear signal. Nuclear/cytoplasmic fractionation is another option to reinforce this idea. Also, why are the cells so subconfluent? The hSMCs show a single cell.

Response: Based on advice from Dr. Haijiang Zhang, Center for Cellular Imaging, University of Gothenburg, we have changed the color of the DAPI signal to violet, which results in a more evident representation of the colocalization now shown in white (Fig. 2A, G). We have also selected the images containing more cells. Furthermore, we have performed nuclear/cytoplasmic fractionation as suggested by the Reviewer. The results of these additional experiments have been added to the revised MS (Fig. 2B, H). The Materials and Methods section has been updated accordingly (page 8).

2. Figure5C: The difference in migration is difficult to see. Considering that loss of STK25 results in impaired SMC proliferation, how can one be sure that the decreased migration seen after 24hrs isn't due to the impaired SMC growth? Alternative migration assays on a shorter time scale would help dissect these two SMC functions.

Response: We thank the Reviewer for this valuable suggestion. We have now performed additional migration assays on a shorter time scale as requested by the Reviewer. The results of these experiments have been added to the Results section of the revised MS (page 14, Fig. 6C). The Materials and Methods section and the figure text have been updated accordingly (pages 7 and 23, respectively).

3. Figure5: While the authors claim that STK25 maintains the SMC phenotype in a contractile state, it would help to show protein expression of the prototypical SMC markers (e.g. SMMHC, SM22a, calponin, etc). What happens to myocardin expression with siSTK25?

Response: We have now measured protein expression of the contractile markers as suggested by the Reviewer. The results of these additional experiments have been added to the Results section of the revised MS as follows (page 14): "Further, we detected markedly higher protein levels of several contractile markers characterizing differentiated smooth muscle cells in STK25-deficient hASMCs [a

several-fold increase in SMMHC (MYH11), SM22 α , and calponin, while no change was observed in myocardin or LMOD1; Supplementary Fig. 5].”

4. Supplementary Figure6: How does the decrease in putrescine in SMCs relate to efferocytosis? (PMID: 32004476)

Response: We would like to thank the Reviewer for bringing this article to our attention. We have now added the following sentences to the Results section of the revised MS (page 15): “In hASMCs, the depletion of STK25 resulted in significantly higher levels of putrescine, which was not observed in STK25-deficient hAECs (Supplementary Fig. 8). Of note, it has recently been demonstrated that in macrophages putrescine augments the clearance, or efferocytosis, of apoptotic cells, ultimately contributing to the resolution of atherosclerosis.⁴⁹ The capacity of engulfment of apoptotic cells has also been reported in vascular smooth muscle cells;⁵⁰ however, it is not known whether putrescine mediates efferocytosis in this cell type.”

5. The authors present data to indicate cell-autonomous differences in vascular cell types (SMC verses EC) and would benefit from further discussion. How does a ubiquitously expressed kinase mediate unique effects depending on cell type?

Response: We thank the Reviewer for this important comment. We have now discussed the cell-type specific versus non-cell-type-specific functions of STK25 in the Discussion section of the revised MS as follows (pages 18-19): “In addition to reduced lipid accumulation in STK25-deficient endothelial and smooth muscle cells reported in this study, we have previously shown that antagonizing STK25 signaling also mitigates ectopic fat storage in liver, skeletal muscle, and kidney.^{9, 10, 12, 14, 16, 18} Of note, in spite of this similar functional readout, the mode of action of STK25 in the regulation of cellular lipid deposition appears to be different in different tissues. To this end, we found that the abundance of acetyl-CoA carboxylase (ACC), which suppresses lipid oxidation and stimulates lipid synthesis,⁶³ is decreased in liver cells where STK25 is knocked down^{10, 12, 16, 20} but not in STK25-deficient endothelial or smooth muscle cells (Supplementary Fig. 10). Interestingly, in contrast to the similar impact on intracellular fat storage in several tissues, multiple functional effects of STK25 vary across diverse cell types. As an example, we here found that silencing of STK25 robustly reduced the proliferation rate and calcification in smooth muscle but not in endothelial cells. The mechanisms behind these cell-type dependent roles of STK25 remain unknown; however, it is likely that specificity is achieved via the interactions of ubiquitously expressed STK25 with different tissue-specific partner proteins.”

6. It is unclear as to why a phospho-proteomic analyses were not performed for ECs in parallel. The manuscript is clearly laid out as a comparison of vascular cell types to better understand STK25 function. The inclusion of an endothelial screen would have greatly enhanced the paper as it would present plausible reasons to address the previous question. It is possible, however, that finances were a factor in study design. If indeed the case, then a follow up of the current SMC screen should be included to strengthen the findings of an overall descriptive paper.

Response: As the Reviewer points out, we had to focus the global phosphoproteomic analysis on one cell type only and we selected hASMCs, where the effect of STK25 depletion was relatively more pronounced. We have now followed up of the global proteomic screen carried out in hASMCs by enrichment analysis of subsystems (pathways) and reporter metabolite sets by integrating the proteomic data obtained from STK25-deficient hASMCs with the genome-scale metabolic model Human1 (Robinson et al, Sci Signal, 2020). The results of this analysis have been added to the Results section of the revised MS as follows (page 17): “Enrichment analysis performed by integrating the proteomic data obtained from STK25-deficient hASMCs with the genome-scale metabolic model Human1⁶¹ revealed changes in cellular dynamics both in terms of subsystems and reporter metabolite sets, which could potentially be affected by STK25 depletion. This analysis demonstrated that STK25 knockdown associated with activation of fatty acid metabolism and oxidation subsystems and, in specific, with upregulation of β -oxidation pathways (Supplementary Fig. 9A), which is well aligned with the enhanced β -oxidation rate detected in STK25-deficient hASMCs using palmitic acid as

substrate (Fig. 1E). In addition, the enrichment analysis suggested co-upregulation of amino acid metabolism pathways by silencing of STK25, implying close relations to fatty acid metabolism (Supplementary Fig. 9A). The elevations of these subsystems were coherently reflected by the enrichment in sets of reporter metabolites representing various forms of CoA, such as acetyl-CoA, which is the hub metabolite of lipid metabolism (Supplementary Fig. 9B).” The Materials and Methods section of the revised MS has been updated accordingly (page 11).

Minor Comments:

1. The manuscript refers to Figure 4A prior to mention of Figure 3H.

Response: We thank the Reviewer for pointing this out. We now present the analysis of oxidative and ER stress markers in aortic root sections from Stk25 knockout and transgenic mice (previously Fig. 3H-I and Fig. 4D-E) as a separate figure (new Fig. 5).

Reviewer #3: The manuscript entitled “Silencing of STE20-Type Kinase STK25 in human aortic cells is atheroprotective” by Cansby et al. demonstrates an atheroprotective role of STK25 deletion in human aortic SMC and EC primary cultures. Specifically, the authors show that STK25 deletion in HASMC and HAEC reduces lipid deposition accompanied with reduced inflammatory and fibrotic markers as well as lowered oxidative and ER stress. This is a well-written and well-packaged manuscript pertinent to pathogenesis of atherosclerosis. Earlier work by the group demonstrated an atheroprotective outcome of STK25 deletion in vivo; in the current manuscript, the authors extended those findings to two key vascular cell types (VSMC, EC) to delineate cell-specific regulatory role of STK25 in atherosclerosis. While the manuscript reveals interesting results, novelty is somewhat limited in the light of their earlier publication (ATVB 2018). However, a few additional experiments (noted below) would enhance the strength of this manuscript. Nevertheless, this is an interesting paper, well-articulated with excellent presentation of data.

Response: We thank the Reviewer for interest in our MS and for very valuable comments. We have carefully revised the MS according to these suggestions and below we have provided a point-by-point response to all comments.

Specific concerns

1. In vitro studies presented have solely utilized siRNA gene silencing in HASMC and HAEC. Manuscript would strengthen if findings can be confirmed in murine aortic SMC and EC from STK25 KO mice for cross-species validation. Such data would also provide direct evidence for a cell-autonomous mechanism underlying the atheroprotective effect of STK25 deletion reported by the authors earlier (ATVB 2018). I suggest that you repeat a few experiments (eg. Figs 1, 2, 5A, B) in murine cells from the STK25 KO mice.

Response: We thank the Reviewer for this valuable suggestion. Unfortunately, no cohorts of Stk25^{-/-} and wild-type mice for harvesting these samples are currently available. For cross-species validation, we have instead used commercially available primary mouse ASMCs and mouse AECs transfected with Stk25-specific siRNA or with a non-targeting control siRNA. The results of these additional experiments have been added to the Results section of the revised MS as follows (page 14): “Notably, we also detected suppressed lipid storage as well as lower levels of oxidative and ER stress markers in primary mASMCs and mAECs transfected with Stk25 siRNA vs. NTC siRNA after metabolic challenge, providing further support for a cell-autonomous mechanism underlying the atheroprotective effect observed in Stk25^{-/-} mice (Supplementary Fig. 3A-D).” The Materials and Methods section of the revised MS has been updated accordingly (page 4).

2. How does STK25 deletion affect SMC proliferation in vivo? Cells in culture are known to behave differently from in vivo. Additional experiments using reporter mice combined with STK25 KO in vivo would greatly strengthen findings and confirm regulatory role of STK25 on VSMC phenotypic switching in vivo.

*Response: We agree with the Reviewer that it would be very valuable to compare smooth muscle cell proliferation in vivo using tissue-specific *Stk25*^{-/-} mice and wild-type littermates. However, performing this investigation would require about 18-24 months of additional studies (including generation of the models, breeding program, and phenotypic characterization) and we respectfully submit that this is beyond the scope of this 3 months revision.*

3. In Fig 5, how does STK25 affect Runx2, Spp1 at protein level. How about SMC contractile markers (Myh11, LMOD-1). what is the molecular mechanism by which STK25 regulates VSMC phenotypic switching? Additional experiments are required to delineate this aspect.

Response: We thank the Reviewer for this important comment. We have now measured protein expression of RUNX2 and SPP1 as well as the contractile markers as suggested by the Reviewer. The results of these additional experiments have been added to the Results section of the revised MS as follows (pages 14-15): “Further, we detected markedly higher protein levels of several contractile markers characterizing differentiated smooth muscle cells in STK25-deficient hASMCs [a several-fold increase in SMMHC (MYH11), SM22 α , and calponin, while no change was observed in myocardin or LMOD1; Supplementary Fig. 5].” and “We also detected lower mRNA expression of the calcification markers osteopontin (SPP1) and RUNX2 in STK25-deficient hASMCs (Fig. 6E), which was accompanied by a tendency for reduced protein abundance (Fig 6F).”

4. Fig. 3H, 3I, 4D, 4E: mark lumen side in images. Also provide corresponding H&E and specificity controls for IHC images.

Response: We have now marked lumen side and provided specificity controls for these images as suggested by the Reviewer (Fig. 5A-D and Supplementary Fig. 11A-B, respectively). Negative controls by excluding primary antibody (Supplementary Fig. 11A) or by substituting a primary antibody by an equivalent concentration of the corresponding Ig isotype (Supplementary Fig. 11B) did not detect any positive signal, confirming the specificity of the immunostaining. We were not able to add corresponding H&E-stained images since no consecutive aortic sinus sections were available from these mice.

Thank you once again for facilitating the review of our MS.

Sincerely,

Prof. Margit Mahlapuu

REVIEWERS' COMMENTS:

Reviewer #1 (Remarks to the Author):

The authors have done a good job with the revised manuscript. I have no further queries.

Reviewer #2 (Remarks to the Author):

The authors have addressed all previous comments, providing additional experiments where necessary. This paper is overall an interesting paper, as it now builds on previous in vivo observations with a mechanistic angle from a cell autonomous perspective. The data is well-presented where key findings begin to draw questions toward the cell-type specific role of STK25.

Reviewer #3 (Remarks to the Author):

This is revised version of the manuscript entitled "Silencing of STE20-Type Kinase STK25 in human aortic cells is atheroprotective" by Cansby et al. In this manuscript, the authors have expanded and complemented their previous report demonstrating an atheroprotective outcome of Stk25 deletion as it applies to VSMC and EC cultures. The overarching goal is to delineate the cell-specific regulatory role of Stk25 in atherosclerosis, with a focus on EC and VSMC.

In this revised version, the authors have significantly improved the manuscript and have satisfactorily addressed my earlier concerns.

I do not have any additional questions or concerns at this time.